# Analysis and Modeling of Sunscreen Ingredients' Behavior in an Aquatic Environment

**Gema Ruiz-Gutiérrez [1], Araceli Rodríguez-Romero [2], Antonio Tovar-Sánchez [3] and Javier R. Viguri Fuente [1,\*]**

1. Green Engineering & Resources Research Group (GER), Departamento de Química e Ingeniería de Procesos y Recursos, ETSIIT, Universidad de Cantabria, Avda. de los Castros 46, 39005 Santander, Cantabria, Spain
2. Departamento de Química Analítica, Facultad de Ciencias del Mar y Ambientales, Universidad de Cádiz, Campus Universitario Río San Pedro, 11519 Puerto Real, Cadiz, Spain
3. Departamento de Ecología y Gestión Costera, Instituto de Ciencias Marinas de Andalucía (CSIC), Campus Universitario Río San Pedro, 11519 Puerto Real, Cadiz, Spain
* Correspondence: vigurij@unican.es

**Abstract:** Sunscreens have become a product based on increasingly complex formulations that include, among many ingredients, a mixture of UV filters to provide optimal sun ultraviolet radiation protection. A significant group of scientific works deals with the impact of UV filters in aquatic media. However, the knowledge of the mechanism and kinetics of the compound's direct release, fate, and its transformation and interaction with living organisms is necessary to assess its environmental occurrence and behavior and to predict potential and real impacts on the aquatic environment. This review outlines the existing analysis and modeling of the release and behavior of sunscreen's ingredients in the marine environment, including aquatic organisms. The physical-chemical properties, photodegradation, and release kinetics of particles and chemicals into the water are studied by hydrodynamic and kinetic models. Direct photolysis of chemicals is modeled as pseudo-first-order kinetics, while the indirect pathway by the reaction of sunscreen with reactive oxygen species is described as second-order kinetics. The interaction of UV filters with marine biota is studied mainly by toxicokinetic models, which predict their bio-accumulation in the organisms' tissues. These models consider the chemicals' uptake and excretion, as well as their transfer between different internal animal organs, as a first-order kinetic process. The studies analyzed in the present work represent a driver of change for the beauty and personal care industry, in order to seek new ecological alternatives through the application of R&D tactics.

**Keywords:** sunscreen; chemicals; modeling; behavior; aquatic organism; accumulation

## 1. Introduction

The growing awareness of beauty and personal care has increased in recent years, reaching historical growth in the products used in this area manufactured by the organic fine chemical sector. Despite the fact that theglobal cosmetic market decreased in 2020 by eight percent, in comparison to the previous year, as a consequence of the COVID-19 pandemic, skincare was the leading category of cosmetic products accounting for about 42 % of this global market. Skincare is one of the most profitable products, as its revenue is projected to generate roughly USD 177 billion in 2025. Skincare encompass a large number of products, such as exfoliants, mineral-based facial treatments, serums, oils, or sunscreens. Solar protection is one of the key drivers behind the fast-growing suncare global market, which generated a revenue of USD 9.3 billion in 2020. The global market for sunscreen ingredients is forecasted to grow from USD 691.6 million in 2021 to USD 787.0 million by 2026, at a compound annual growth rate of 2.6% for the 2021–2026 period [1,2].

Sunscreens are among the many emerging pollutants that enter the sea and cause adverse ecological effects [3,4]. In recent years, the environmental effects of some

ingredients used in the preparation of sunscreens once they are released into the aquatic environment has been discussed by the scientific community [5,6]. The increase in tourism associated with the sun and coast has increased the presence of these products in the marine environment and, consequently, the concern about the potential dangers to marine ecology, which are mainly caused by the UV chemical filters, both organic and inorganic, present in the sunscreens [5,7,8]. Following the verification of these adverse effects on marine life and ecosystems, specific ingredients used in these products have been banned in some countries and regions, mostly coastal and intensively touristic areas (e.g., Hawaii, Palau, and Thailand).

The analysis and modeling of the release, behavior, and fate of ingredients contained in sunscreen products when they come into contact with seawater allow us to predict their potential negative impacts on the marine environment. The development of a general modeling method, which allows us to model these complex (from an environmental point of view) matrices based on chemical cocktails, requires the minimization and management of the effects caused by the use of sunscreens in aquatic ecosystems. In the present work, we first present a brief perspective on sunscreen products. Secondly, we critically review our current knowledge of models studied in the literature regarding the behavior and transformation of these environmental harmful substances and their associated ingredients in aquatic media. This review may represent an advance in the current state of knowledge, given that it adds a different point of view on sunscreen release in aquatic ecosystems, investigating the chemical behaviour of sunscreen compounds depending on the experiments performed (in-lab, in-field, with organisms, without organisms). The study also reports the mechanisms and kinetic models used, as well as the requirement for the more precise and realistic modeling of skincare ingredients and, in general, for all the emerging contaminants.

### 1.1. Sunscreens: Function, Market and Legislation

Sunscreens are cosmetic products belonging to the skincare products group. Their main function is to protect individuals, in both short and long terms, from solar UV radiation that the human skin receives. These products, primarily designed to be used in high-UV-radiation-exposure situations, have evolved into multifunctional products capable of preventing skin aging due to their moisturizing effect [9].

The UV filter market dominates the sector, as UV filters are an ingredient used in cosmetic products as well as being used exclusively in the formulation of sunscreen. Higher-UV-protection products are demanded by both sectors, being a blend of UV filters the basic ingredients that provide this protection. From a global market volume of 27,000 tons of UV filters in 2016, an annual growth rate of 4% is expected [10].

The complex formulations of sunscreens, their use, and the possible release into the environment of the ingredients that make up their photodegradation and aging properties make it necessary to develop and apply some type of legislation or regulation to both the manufacturing and use of these products. In recent years, legislation proposals have been developed for these components, but not all of them have been enacted due to the lack of adequate research that demonstrates their harmful effects on the environment. Several governments (e.g., the states of Hawaii and Florida (Key West), US Virgin Islands, Palau, and Bonaire) have approved legislations banning the sale of sunscreens containing some UV organic filters, such as oxybenzone and octinoxate, due to their reported effects on aqueous environments and potential for bleaching coral reefs. Additionally, a recent study [11] conducted in the Mexican Caribbean highlighted the need to create more restrictive regulations on sunscreens. Table 1 includes IUPAC chemical name, chemical structure, regulation, and main properties of the most commonly used UV filters, including 10 organic and 2 inorganic compounds.

**Table 1.** IUPAC chemical name, chemical structure, regulation, and main properties of the most common UV filters used in sunscreens. Red color represents banned chemicals.

| UV Filter | IUPAC Chemical Name (Formula) | Chemical Structure | Regulation(*) (**) | Properties |
|---|---|---|---|---|
| Benzophenone -3 Oxybenzone (BP3, BZ-3) | 2-Hydroxy-4-methoxybenzophenone ($C_{14}H_{12}O_3$) |  | Regulated by maximum contents in the US, Canada, the EU, ASEAN, MERCOSUR, China, India, Japan, Korea, Australia, NZ, and South Africa. Banned in Palau, Thailand marine natural parks, Aruba, Bonaire, the US Virgin Islands, Key West (Florida, US), and Hawaii (US). | CAS number: 131-57-7; molecular weight (MW): 228.24 Density: 1.201 g/cm³ Solubility in water: <0.1 g/100 mL at 20 °C pK$_a$ = 7.56; log K$_{ow}$ = 3.52 |
| Benzophenone-4; Sulisobenzone (BP4) | 5-Benzoyl-4-hydroxy-2-methoxybenzene-1-sulfonic acid ($C_{14}H_{12}O_6S$) |  | Regulated by maximum contents in the US, Canada, the EU, ASEAN, MERCOSUR, China, India, Japan, Korea, Australia, NZ, and South Africa. | CAS number: 4065-45-6; MW: 308.31 Solubility in water: 1 g per 4 mL pK$_{a1}$ = −0.70 (sulfonic acid); pK$_{a2}$ = 7.56 (hydroxyl); log K$_{ow}$ = 0.37 |
| Octinoxate; Uvinul MC80; Octyl methoxycinnamate (OMC) | (RS)-2-Ethylhexyl (2E)-3-(4-methoxyphenyl) prop-2-enoate ($C_{18}H_{26}O_3$) |  | Regulated by maximum contents in the US, Canada, the EU, ASEAN, MERCOSUR, China, India, Japan, Korea, Australia, NZ, and South Africa. Banned in Palau, Thailand marine natural parks, US Virgin Islands, Key West (Florida), and Hawaii (US). | CAS number: 5466-77-3; MW: 290.40 Density: 1.010 g/cm³ Insoluble in water log K$_{ow}$ = 5.80 |
| Octocrylene; Uvinul N-539 (OC) | 2-Ethylhexyl 2-cyano-3,3-diphenylprop-2-enoate ($C_{24}H_{27}NO_2$) |  | Regulated by maximum contents in the US, Canada, the EU, ASEAN, MERCOSUR, China, India, Japan, Korea, and South Africa. Banned in Palau and US Virgin Islands. | CAS number: 6197-30-4; MW: 361.48 Density: 1.055 g/cm³ Insoluble in water log K$_{ow}$ = 7.35 |
| Enzacamene; 4-Methylbenzylidene camphor (4-MBC) | (3E)-1,7,7-Trimethyl-3-[(4-methylphenyl) methylene]-2-norbornanone ($C_{18}H_{22}O$) |  | Regulated by maximum contents in Canada, the EU, ASEAN, MERCOSUR, China, India, Japan, Korea, Australia, NZ, and South Africa. Banned in Palau and Thailand marine natural parks. | CAS number: 36861-47-9; MW: 254.37 Density: 1.064 g/cm³ Insoluble in water log K$_{ow}$ =5.47 |

| | | | Regulation | Properties |
|---|---|---|---|---|
| Ácido p-aminobenzoico (PABA) | Ácido 4-aminobenzoico ($C_7H_7NO_2$) | | Regulated by maximum contents in the US, Canada, the EU, ASEAN, MERCOSUR, China, India, Japan, Korea, Australia, NZ, and South Africa. | CAS number: 150-13-0; MW: 137.14 Density: 1.374 g/cm³ Solubility in water: 1 g/170 mL (25 °C) $pK_{a1} = 2.38$; $pK_{a2} = 4.85$; log $K_{ow} = 0.83$ |
| Padimate O; Escalol 507; octyldimethyl PABA (OD-PABA) | 2-ethylhexyl 4-(dimethylamino) benzoate ($C_{17}H_{27}NO_2$) | | Permitted UV filters and regulated by maximum contents in the US, Canada, the EU, ASEAN, MERCOSUR, China, India, Japan, Korea, Australia, NZ, and South Africa. | CAS number: 21245-02-3; MW: 277.40 Density: 0.990 g/cm³ Solubility in water: 0.54 mg/L $pK_a = 2.9$; log $K_{ow} = 5.77$ |
| Ensulizole; phenyl benzimidazole sulfonic acid (PBSA) | 2-Phenyl-3H-benzimidazole-5-sulfonic acid ($C_{13}H_{10}N_2O_3S$) | | Regulated by maximum contents in the US, Canada, the EU, ASEAN, MERCOSUR, China, India, Japan, Korea, Australia, NZ, and South Africa. | CAS Number: 27503-81-7 Molecular M.: 274.29 Soluble in water |
| 2,4-Dihydroxy Benzophenone; Benzophenone-1 (DH-BP) | (2,4-dihydroxyphenyl)phenyl-methanone ($C_{13}H_{10}O_3$) | | Regulated by maximum contents in Japan and South Africa. | CAS number: 131-56-6; MW: 214.22 Density: 1.302 g/cm³ Insoluble in water; log $K_{ow} = 3.17$ |
| Dioxybenzone; Benzophenone-8 (DHM-BP) | 2,2′-Dihydroxy-4-methoxybenzophenone ($C_{14}H_{12}O_4$) | | Regulated by maximum contents in the US, Canada, ASEAN, MERCOSUR, Korea, Australia, NZ, and South Africa. | CAS number: 131-53-3; MW: 244.24 Density: 1.38 g/cm³ Insoluble in water; $pK_a = 7.11$; log $K_{ow} = 4.31$ |
| TiO₂ nanoparticle | Titanium dioxide (TiO₂) | | Regulated by maximum contents in the US, Canada, the EU, ASEAN, MERCOSUR, China, Japan, Korea, Australia, and NZ. | CAS number: 13463-67-7 XRD: rutile, anatase Al(OH)₃ or SiO₂ coating; TiO₂ %: 79–89 |
| ZnO nanoparticle | Zinc oxide (ZnO) | | Regulated by maximum contents in the US, Canada, ASEAN, MERCOSUR, China, Japan, Korea, and NZ. | CAS number: 1314-13-2; XRD: wurtzite SiO₂-coated or coated with a silicone derivative. ZnO %: 79.1–81.5 |

(*) In the Mexican states of Quintana Roo and Yucatán, biodegradable sunscreens are permitted and the use of sunscreen in sinkholes is regulated in several municipalities. (**) ASEAN (Brunei, Cambodia, Indonesia, Laos, Malaysia, Myanmar, Philippines, Singapore, Thailand, and Vietnam); MERCOSUR (Argentina, Brazil, Paraguay, and Uruguay).

*1.2. Design and Manufacturing of Sunscreen*

There is a large variety of possible UV filters to be used in the formulation of a sunscreen. These are divided into two groups: organic (synthetic) and inorganic (minerals). Although the formulation of each product is different, all of them combine several UV filters, whether organic or inorganic. This is because the maximum concentrations allowed are lower than those that a protector would require to provide effective protection against UV radiation. In addition, by using different types of UV filters, a broader spectrum of protection is achieved [12]. The most common UVA organic filters are benzophenones 1, 2, 3, and 4, which are aromatic ketones only effective against short-wave UVA radiation. Benzophenones 4, 5, and 9 have sulfur and sodium atoms in their formulation. Ecamsule, avobenzone, and meradimate (with nitrogen atoms included in their compositions) are other commonly used filters. An important group of UVB filters are cinnamates (found in 90% of sunscreens). Octyl methoxycinnamate (OCM) is widely used, but presents an volatility to sun exposure. p-aminobenzoic acids (PABAs) and derivatives, salicylates, camphor derivatives, octocrylene, ensulizol, and urocanic acid are other groups of important UVB filters.

Inorganic filters used in sunscreen formulations are titanium dioxide and zinc oxide and, in turn, can be included as nanoparticles. The photocatalytic properties of nanoparticles cause their photopassivation in order to prevent any type of oxidative stress on the skin or the product itself from losin its protective characteristics [13,14]. In general, this photopassivation is achieved by the precipitation of an inert mineral layer of aluminum hydroxide or silica on the surface of the nanoparticles [13]. Another way to increase the photostability of $TiO_2$ nanoparticles ($TiO_2$-NPs) is by "doping" the particle with certain metals, such as vanadium, manganese, iron, chromium, or cobalt [15].

Sunscreen is a product classified within the organic fine chemical and pharmaceutical industry where manufacturers produce a wide range of chemical substances, which are typically of a high added value. The most important task in their manufacturing process by recipe-driven batch processes in multipurpose plants is to achieve compliance with all the desired functional characteristics and fulfil the requirements for the approval of the product by different administrations.

The different combinations of the formulated ingredients focus on obtaining a specific sun-protection factor, but also on meeting the needs of a specific group of consumers; in this sense, cream, lotion, gel, stick, or aerosol media allow different forms of application for different people. All of this gives rise to a complex matrix where, in addition to UV-filter combinations, aqueous and organic dispersing agents, emulsifiers, antioxidants, humectants, preservatives, stabilizers, and antimicrobial agents are included in the formulations.

When sunscreens are used, part of the many ingredients used in the sunscreen's formulation are released into the surrounding environment, especially into the marine environment [8]. This release depends on the sunscreen ingredients' properties, the receiving medium characteristics, and a series of physical–chemical interactions between both of them [16–20]. All these processes can cause the ingredients to remain unchanged in the medium, change their speciation, or even transform into other compounds at variable rates [21–26]. Once in the environment, the original or transformed ingredients can interact directly or indirectly with the environment and marine life, causing a series of adverse effects [27–32]. Different mechanisms and mathematical models are proposed in the present study to analyze and predict the compound's release, its transformation, and interaction with the environment and living organisms, e.g. [33–38].

## 2. Analysis and Modeling of the Behavior of Sunscreen's Ingredients in an Aquatic Environment in the Absence of Organisms

The behavior, transformation, aging, and modeling of sunscreen's ingredients in the aquatic environment in the absence of organisms were identified through a systematic literature review. Advanced oxidation processes (AOPs), that is, processes that were not spontaneous in nature, were not considered. Relevant keywords were combined to form a broad search conducted through the ScienceDirect and Scopus databases. The search was limited to the papers published in the year 2000 and onwards. This process identified 25 published articles that fulfiled the objective of this section. Figure 1 classifies the publications founded according to the scale and ingredients studied. Table 2 summarizes the modeling approach used in the different studies reviewed in Section 2.

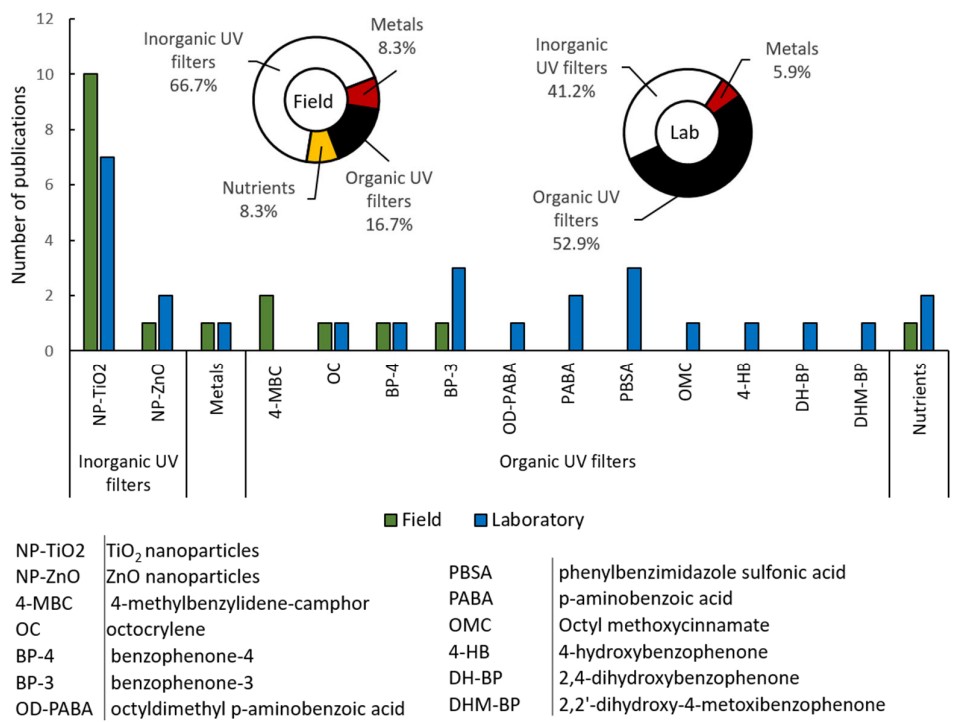

| NP-TiO2 | TiO$_2$ nanoparticles | | PBSA | phenylbenzimidazole sulfonic acid |
|---|---|---|---|---|
| NP-ZnO | ZnO nanoparticles | | PABA | p-aminobenzoic acid |
| 4-MBC | 4-methylbenzylidene-camphor | | OMC | Octyl methoxycinnamate |
| OC | octocrylene | | 4-HB | 4-hydroxybenzophenone |
| BP-4 | benzophenone-4 | | DH-BP | 2,4-dihydroxybenzophenone |
| BP-3 | benzophenone-3 | | DHM-BP | 2,2'-dihydroxy-4-metoxibenzophenone |
| OD-PABA | octyldimethyl p-aminobenzoic acid | | | |

**Figure 1.** Classification of publications based on the components of sunscreens, grouping them into organic and inorganic filters, metals and nutrients, and split between field and laboratory studies. ▮ Field studies; ▮ laboratory studies.

### 2.1. Modeling of the Sunscreen Ingredients' Behavior in Field Scenarios

A large number of studies focus on the concentration and fate of sunscreen ingredients in different environment scenarios that are commonly used for recreational activities, such as rivers, lakes, estuaries, and beaches [8]. Substance or material flow analysis (SFA or MFA); particle flow analysis (PFA); and stochastic Lagrangian and hydrodynamic models, which are sometimes conducted together, have been previously used to model the behavior of sunscreen UV filters in the field [11,34,37,39–43].

Material flow analysis models describe the accumulation of the studied compound in different compartments, both productive and environmental, considering each compartment as a mixer (MFA models) or considering the probabilistic behaviors (PMFA models) [35–41,43]. Hydrodynamic models (HMs) describe the transport, diffusion, and transformation of the studied compounds in a natural aquatic environment, generally in marine environments, such as a gulf, beach, bay, or an estuary [19,36,40].

Recently, Zheng and Nowack [39] integrated the size distribution of the particle flows into the MFA model by a size-specific, dynamic, probabilistic, MFA (ss-DPMFA) model that computes size information about the $TiO_2$-NPs flows released into the environment. The results obtained from this study could be used as inputs for environmental multimedia fate models.

Lindo-Atichati et al. [40] incorporated the fate and transport of 4-MBC, OC, and BP-4 UV filters into a stochastic Lagrangian model that was coupled with a high-resolution hydrodynamic model; this study generated the expected trajectories of water parcels that transport chemicals between pollution sources (wastewater-treatment plants) and mussel rafts that were exposed to pollution. The predicted concentrations of organic UV filters in the coastline and at the outfalls of urban wastewater-treatment plants were provided.

Arvidsson et al. [41], using a global approach, developed a particle-flow-analysis (PFA) methodology to determine the emissions of $TiO_2$-NPs from several industrial products, such as paint, sunscreen, and self-cleaning cement. Their results indicate that the current, highest emissions of $TiO_2$-NPs originate from the use of sunscreen.

The approach of Johnson et al. [19] was a combination of measurements and modeling of the fate of titanium in the influent and effluent in an active sludge plant. Their study calculated river-water concentrations using hydrological models estimating the magnitude and variability of flows across a catchment in combination with a range of water-quality models, including a catchment scale water-quality model. The range of concentrations obtained included other predicted concentration ranges using probabilistic material flow analysis to model the concentrations of nanoparticles in specific environments, including sediment [34–37,42,43]. In addition to these works, the review conducted by Yuan et al. [6] highlighted the lack of information on the environmental fate of inorganic UV filters in the real environment.

### 2.2. Modeling of the Sunscreen Ingredients' Behavior in Lab-Scale Studies

Before analysing the laboratory-scale experiments, we highlighted the works of Tovar-Sánchez et al. [3] and Sánchez-Quiles and Tovar-Sánchez [8], which combined laboratory and field experiments. These works demonstrated that transformations of sunscreen nanoparticles in seawater and under solar radiation produce compounds and reactive oxygen species (ROS) as singlet oxygen ($^1O_2$), hydroxyl radicals (OH), and superoxide radicals ($O_2^-$). They suggested that sunscreens in coastal waters may have deleterious effects on the coastal ecosystem, either by inhibiting the growth of some species of marine phytoplankton or by adding essential micronutrients (N, P, and Si compounds) that may stimulate the growth of others.

Laboratoryexperiments aim to propose reaction mechanisms and kinetic models to describe the physico-chemical transformations of sunscreen ingredients under controlled conditions. The results of such studies can be inputs to the model applied to real environmental scenarios. In these studies, aqueous solutions of the target chemical were prepared by dissolving individual sunscreen components directly in ultrapure water. Additions of selected concentrations of ions (nitrate, bicarbonate, or chloride) and organic matter are common to examine how their presence may influence the transformations of the studied chemicals [44–48]. In addition to ultrapure water and synthetic seawater, other water matrices, such as freshwater, estuarine, and seawater samples from bathing areas, as well as water from swimming pools, were used as bulk solutions in laboratory-scale experiments [24,31,48–50]. Laboratory experiments were conducted by submitting the liquid mixture to magnetic stirring under simulated sunlight mimicking daily natural cycles using irradiation from laboratory UV lamps. Water evaporated under irradiation was supplemented regularly in some experiments, while controlled temperature representative of seawater was maintained by external cooling in others [5,16,51].

Three sets of laboratory studies can be distinguished: (i) studies that focused on the photodegradation kinetics and mechanism of sunscreen UV organic filters; (ii) works studying the release, fate, transformation, and aging in water of sunscreen ingredients

and their degradation products; and (iii) works that highlighted the mobilization and photoproduction of inorganic nutrients (i.e., $SiO_2$, $P-PO_4^{3-}$, and $N-NO_3^-$) from sunscreen products. These works are analyzed in the following sections in the present work with Figure 2 showing the modeled transformations related to the release and transformation of the sunscreen ingredients in the aquatic environment in laboratory-scale studies.

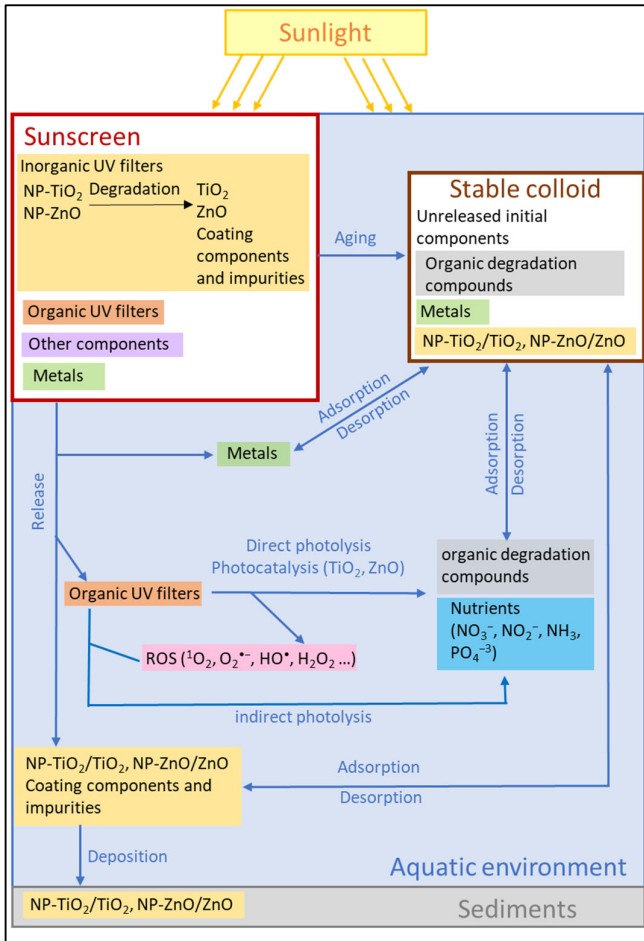

**Figure 2.** Superstructure of the sunscreen's transformations in the environment. References in the figure refer to works that propose process modeling. The most representative references for the process in Figure 2 are: Metals Release and Adsoption/Desorption [5]; Organic UV filters Release [40]; Organic UV filters Direct photolysis [24,25,32,44,45,47,48,50]; Organic UV filters Photocatalysis [52]; Organic UV filters Indirect photolysis [24,44,45,47,50].

### 2.2.1. Photodegradation of Sunscreen UV Organic Filters under Simulated Natural Conditions

Several works [24,25,32,44,45,47,48,50,52] modeled and determined the photodegradation kinetics in laboratory-simulated environmental conditions of organic UV filters as benzophenone-derived compounds (benzophenone-3 (BP-3), benzophenone-4 (BP-4), 4-hydroxybenzophenone (H-BP), 2-hydroxy-4-methoxybenzophenone (HM-BP), 2,4-dihydroxybenzophenone (DH-BP), and 2,2'-dihydroxy-4-methoxybenzophenone (DHM-BP)), and the acid agents p-aminobenzoic acid (PABA), octyl-dimethyl-p-aminobenzoic acid (ODPABA), and 2-phenylbenzimidazole-5-sulfonic acid (PBSA).

Photodegradation processes may occur in water by direct and/or indirect photolysis. The direct photolysis pathway implies self-decomposition after the organic UV filters absorb UV radiation, following a pseudo-first-order kinetics; this model allows a

manageable solution for the kinetic system leading ro an apparent or observed kinetic constant. The indirect photolysis pathway involves different photosensitizer agents, such as dissolved organic matter (DOM), reactive oxygen species (ROS), or other inorganic substances (e.g., nitrates, bicarbonates, NaCl, or $TiO_2$), usually modeled as a second-order reaction rate.

The transformation-rate constant is a strong function of significant environmental parameters, such as pH [45,52], as well as dissolved organic matter, nitrate, and chloride concentrations in water [44,48,53], which slow down the transformation rates of organic UV filters [48]. The above-mentioned works performed experiments with synthetic solutions of sunscreen ingredients, and therefore did not take into account the chemical release process from commercial sunscreens into an aqueous medium. However, the works of Sendra et al. [31] and Tovar-Sánchez et al. [54] studied the photolysis of organic UV filters, starting from solutions of commercial sunscreens in seawater. Although they did not propose reaction models, both works showed the evolution of $H_2O_2$ generation as a consequence of the degradation of both $TiO_2$-NPs and organic filters.

In addition to modeling the UV filters' transformations under simulated natural conditions in thelaboratory, there are works that modeled the degradation of UV filters through the named advanced oxidation processes (AOPs). In these treatment processes, more extreme conditions were used in order to achieve maximum degradation yields. However, the kinetic modeling proposed in these cases can be useful to compare with natural processes [21]. AOP processes for the removal of sunscreen UV filters in water established pseudo-first-order-rate degradation, obtaining an apparent kinetic constant, and second-order-rate constants of the reaction between UV filter and reactive species [49,55–62].

### 2.2.2. Degradation and Aging of $TiO_2$-NPs Used in Sunscreen

Sunscreens consist of a complex oil-in-water or water-in-oil emulsion in which UV filters are incorporated. Titanium dioxide nanoparticles ($TiO_2$-NPs) are usually functionalized with combinations of surface coatings, such as inorganic oxides ($SiO_2$, $Al_2O_3$), polydimethylsyloxane (PDMS), or stearic acid, to satisfy the practical application as an inorganic UV filter being a complex matrix.

Several authors studied the physicochemical processes related to $TiO_2$-NPs degradation and aging in aquatic media during a contact time of up to seven days using aqueous solutions of $TiO_2$-NPs [16,20,53,63] or aqueous solutions of commercial sunscreen containing $TiO_2$-based nanocomposites [51]. The results obtained show that large parts of $TiO_2$-NPs are released from the sunscreen with fast degradation, losing their coating and, therefore, their protective function. The kinetics and thermodynamics of the global release and aging process occurs under a set of processes, with complex and interrelated mechanisms, which depend on the type of $TiO_2$-NPs and the properties of the aquatic environment. $TiO_2$-NP transformations as dissolution, adsorption–desorption, aggregation/agglomeration, sedimentation, sulfidation, mineralization, redox reactions (e.g., photo-oxidation and photoreduction), and interaction with macromolecules can affect its fate, toxicity/bioavailability, and persistence in the environment [7,17,23]. In these processes, the characteristics of $TiO_2$-NPs and the environmental conditions play a crucial role. Properties, such as the crystalline structure, morphology, size distribution, surface charge, type of coating materials and stabilizing agents ($SiO_2$, $Al(OH)_3$, PDMS, and/or stearic acid), must be taken into account, together with the characteristics and conditions of the aqueous medium simulating natural-environment conditions (freshwater, seawater, estuarine water), such as ionic strength; pH; temperature; light; and the presence of ingredients, such as natural organic matter, macromolecules, radicals, and surfactants.

Several works related to the aging and transformation of $TiO_2$-NPs [16,20,51,53,63] proposed the agglomeration of nanoparticles during the aging process in stable colloidal residue that could include macroscopic aggregates, agglomerates, and submicronic fractions; these colloidal residues interact with the organic phase of the sunscreen, and the variety of ingredients in the sunscreen can be released into the aqueous environement.

Metals, elements associated with Ti or Zn nanoparticles, as well as their coatings and metal impurities from raw materials [64–66] can partially dissolve in water and partially remain as nanoparticles associated with organomineral agglomerates and aggregates.

Furthermore, Auffan et al. [63] and Slomberg et al. [16] showed that after 48 h of TiO$_2$-NPs aging in both fresh- and seawater, the protective SiO$_2$ layer was almost completely degraded with the high release of Si; however, Al(OH)$_3$ was more stable with a low release of Al into the water. A review by Peng et al. [67] discussed the environmental behavior of metal-based NPs with an in-depth analysis of the mechanisms and kinetics of aggregation, dissolution, and ROS generation; the review analyzed proposals, such as the Arrhenius-based kinetic model for NP dissolution, interaction force boundary layer theory approaches for describing adsorption, and different diffusion-limited and reaction-limited aggregation models, to predict the aggregation kinetics of NPs in solutions.

### 2.2.3. Inorganic Nutrients of Sunscreen

Organic UV-filter degradation in aqueous media in the presence of TiO$_2$-NPs and light, as well as the degradation of nanoparticle surface coatings, can generate nutrients as nitrogen compounds (NO$_2^-$, NO$_3^-$, NH$_4^+$), phosphorous compounds (PO$_4^{3-}$), and SiO$_2$, which can have a considerable environmental impact [3,5,68].

The photocatalytic degradation of nitrogen-containing compounds from sunscreen can result in nitrate, nitrite, ammonia, and nitrogen production, whose relative concentrations and kinetic processes depend mainly on (i) the nature of nitrogen-containing organic compounds (i.e., the functional group in which nitrogen atoms are presented, the initial oxidation state of nitrogen, the nitrogen-existing type in compounds, and the number and position of nitrogen atoms); (ii) the TiO$_2$-NPs properties as particle-size distribution and/or crystalline structure; and (iii) the reaction conditions, such as the light intensity and duration, concentration of organic compounds, TiO$_2$-NPs concentration, and pH [69,70].

The study conducted by Low et al. [69] presented the formation of ammonia (NH$_3$) with a basic pH or ammonium ion (NH$_4^+$) in acidic conditions; both species transformed into nitrite ions (NO$_2^-$) and, later at much lower rates and following a first-order reaction, into nitrate ions (NO$_3^-$), with chloride and sulphate ions being the inhibitors of the reaction. Jing et al. [70] classified different scenarios according to the types of nitrogen groups in organic compounds of sunscreen: amino groups were predominantly mineralized into NH$_4^+$, while the nitro group was mainly converted into NO$_3^-$. For the heterocyclic structures, the nitrogen atom was transformed into either one or both of the NH$_4^+$ and NO$_3^-$ species. N$_2$ was mainly generated from the photodegradation of the –N=N– double-bond moieties. In relation to the kinetic modeling of the nitrogen compounds' transformation, Jing et al. [70] suggested zero-order kinetics at high concentrations ($5 \times 10^{-3}$ mol L$^{-1}$) because diffusion was the dominant process; in contrast, at lower concentrations, the degradation efficiency was proportional to the substance concentration in accordance with an apparent first-order kinetics.

Moreover, Ji et al. [50] indicated that the sulfonic moiety from 2-phenylbenzimidazole-5-sulfonic acid (PBSA) was primarily converted to SO$_4^{2-}$, while nitrogen atoms were predominantly released as NH$_4^+$ and, to a lesser extent, as NO$_3^-$. Along the same lines, Abdelraheem et al. [61] showed that the mineralization of hetero N-atoms from PBSA and 1H-benzimidazole-2-sulfonic acid (BSA) were released in solution as NH$_4^+$. Ammonia could be oxidized into NO$_3^-$ by hydroxyl radicals generated from visible light/TiO$_2$ present in the aqueous media or resulting from the autocatalyzed photo-oxidation process of NH$_4^+$ by UV/TiO$_2$.

In addition to the nitrogen compounds, organic sunscreen ingredients with phosphorous, as well as phosphorous impurities, from TiO$_2$-NPs, can release species of phosphorous as inorganic nutrients into the aquatic environment with a probably significant negative environmental impact. Liu et al. [68] studied the phosphorus-release kinetics from five types of TiO$_2$-NPs under the influence of varying solution chemistries. Similar to the

previous literature, the authors reported a rapid, initial decline (2 h) in the amount of ad-sorbed phosphate on the surface, followed by an extended slow desorption rate, and after phosphorous dissolution, re-adsorption on the particulate surface was reported. This interaction between aqueous $PO_4^{3-}$ and nanoparticle surfaces (ZnO-NPs) was studied by Lv et al. [71], who showed the dissolved $Zn^{2+}$ formed nanoparticles of zinc (ZnO-NPs) complexed with aqueous $PO_4^{3-}$, resulting in the formation of amorphous zinc phosphate hydrate precipitate. In both the studies previously mentioned, the introduction of phosphorus into the aquatic environment increased the concentration of nutrients, in addition to being bioavailable, generating an impact on the environment.

Similar to nitrogen and phosphorus, silicon compounds can release species containing this element, constituting a new nutrient element in the medium. Liu et al. [72] investigated the release kinetics of Si impurity frequently introduced during NP synthesis, and the resulting effect on $TiO_2$-NPs transformation in aqueous solutions. The Si-release kinetics follow the parabolic kinetics mode, which is similar to the diffusion-controlled dissolution of minerals. Slomberg et al. [16] evaluated the transformation of the UV $TiO_2$-NPs filter with a surface $SiO_2$ coating, showing a high level of $SiO_2$-layer dissolution after 48 h in fresh and marine waters.

On the other hand, Rodríguez-Romero et al. [5] studied the combined release of metals and nutrients from a commercial sunscreen. A pseudo-first-order kinetics model was proposed to describe the metal-release process, considering that metals are released from the organic phase of the sunscreen into the aquatic environment and can then be adsorbed on the stable colloid formed by the aging of the initial sunscreen. The photodegradation of organic compounds from sunscreen with phosphorus and silica, releases phosphates and silica into the seawater, which is described as first-order kinetics.

**Table 2.** Models and the modeling approach used in the different reviewed studies described in Section 2.

| Model | Modeling Approach and Model Expressions | Chemical | Ref. |
|---|---|---|---|
| **Sunscreen ingredients' modeling in field scenarios** | | | |
| Substance flow analysis (SFA), material flow analysis (MFA), probabilistic and dynamic probabilistic MFA (ss-DPMFA). Particle-flow analysis (PFA). | • Mass-balance multi-compartment models.<br>• Particle number is used as a flow and stock metric instead of mass.<br>• Probabilistic material flow analysis to model nanoparticle concentration in compartments. | $TiO_2$-NPs | [35–41,43,19,36] |
| Hydrodynamic models (HMs). | Hydrodynamic numerical models describing and estimating the velocity field of three-dimensional currents and turbulent vertical mixing. Combination with water-quality models. | $TiO_2$-NPs, 4-MBC, OC, BP4 | [19,40] |
| **Kinetic model of laboratory-scale photodegradation of sunscreen UV organic filters** | | | |
| ▪ Direct photolysis: pseudo-first-order.<br>▪ Indirect photolysis: second order.<br>▪ Langmuir–Hinshelwood model. | $r_{i,j} = k_{i,j}\, C_i$<br>$r_{i,j} = k_{i,j}\, C_i^{2}$<br>$r_{i,j} = \dfrac{k_{i,j}\, K\, C_i}{1 + K\, C_i}$ | BP3, BP4, H-BP, HM-BP, DH-BP, DHM-BP, PABA, OD-PAB, PBSA, 4-MBC | [24,25,32,44,45,47,48,50,52] |
| **Degradation and aging of $TiO_2$-NPs used in sunscreen** | | | |
| Several kinetic models for NP dissolution coupled with adsorption and aggregation. | Pseudo-first- and second-dissolution kinetic orders. Diffusion or reaction limited agregation models. | $TiO_2$-NPs | [16,20,51,63,67] |
| **Inorganic nutrients from sunscreen** | | | |
| Pseudo-first-order kinetic model. Zero-order kinetics at high nitrogen concentrations. | $r_{i,j} = k_{i,j}\, C_i$<br>$r_{i,j} = k_{i,j}$ | Organic UV filters with $N_2$ and P compounds. P, $SiO_2$ impurities/coatings. | [5,50,68–72] |

### 3. Analysis and Modeling of the Behavior of Sunscreen Ingredients in Contact with Aquatic Organisms

The mechanism and mathematical models proposed to analyze and predict the compound's release, transformation, and interaction with living aquatic organisms were identified through a systematic literature review. Figure 3 presents the number of published articles that fulfil the objective of this section, collected according to the type of substance (UV filters or metals), differentiating between field (in green) and laboratory studies (in blue). It also presents the number of publications according to the aquatic organism studied. The most studied organisms were bivalves, which are good biomonitors of coastal pollution because they are filter feeders and exposed to high volumes of water for feeding and respiration [73,74]. Furthermore, their commercial and ecological importance in coastal environments make bivalves an excellent potential target species for assessing the effects of marine pollution [75].

The research conducted on metal accumulation is more extensive than for UV filters (Figure 3). High concentrations of metals in the marine environment near industrialized areas have led to an interest in their bioaccumulation and possible influence on the food chain, as well as the availability of tools for determining their effects. However, existing studies with UV filters have focused more on determining the toxic effects on aquatic organisms [67,76–80].

Due to the increasing use of sunscreens, their chemical components have reached the aquatic environment by direct sources due to various leisure and recreational activities (e.g., swimming, diving and wave-riding, and/or indirect sources through effluents from wastewater-treatment plants [81–83]). Once in the aquatic environment, these compounds can bioaccumulate in living organisms. Furthermore, this bioaccumulation can lead to biomagnifications where species higher in the food chain may be exposed to all the chemicals that lower-order species accumulate [84].

Organic UV filters are expected to accumulate in aquatic biota due to their lipophilic character and solubility in water. Many organic UV filters have high lipophilicity, which makes them susceptible to accumulate in the organisms of the biota, being stored somewhat faster than they are metabolized or excreted, thus primarily fixing them in soil and sediments. These matrices are considered a sink for organic pollutants in aquatic ecosystems [27].

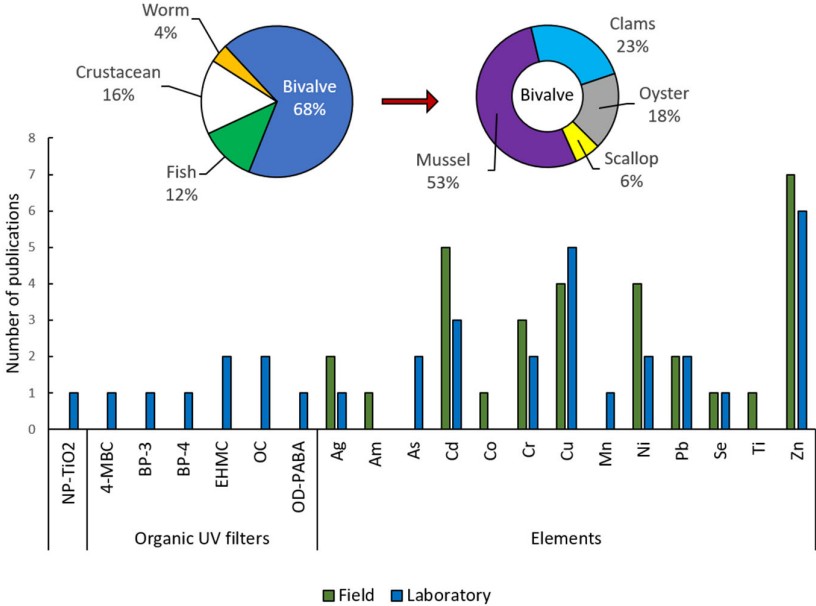

**Figure 3.** Classification of publications based on the components of sunscreens (UV filters and metals), according to the aquatic organisms studied and splitting between field and laboratory studies. ▇ Field studies; ▇ laboratory studies.

Additionally, inorganic filters cause environmental toxicity and enzymatic changes in aquatic organisms. Several studies identified and evaluated the toxicity and ecotoxicological effects (different biological responses) of $TiO_2$ and ZnO nanoparticles on different aquatic species [28,30,76,77,85]. Both of them can bioaccumulate and cause oxidative stress in clams [31,77].

In general, the resaerch collected from the literature study the harmful effects on living organisms of UV filters (only one filter or mixtures of several filters prepared in the laboratory). Only a few studies on the effects of sunscreens on marine organisms have been conducted using commercial sunscreen products, e.g., [29,79,86,87].

In environmental studies, the modeling itself is used for quantification, synthesis, and prediction (or hypothesis testing). Bio-toxicokinetic models, based on the description of the bioconcentration rate of chemical substances in living organisms, play an important role in driving the field and provide a greater understanding of the bioaccumulation and toxicity of chemical compounds in aquatic organisms. Therefore, there have been many attempts to develop bio-toxicokinetic models to predict the bioaccumulation, biotransformation, and toxicity of metals in aquatic organisms, the studies dealing with organic compounds being scarce, such as organic UV filters. This is probably because these organic compounds in organisms can be digested and transformed into different ones, which in some cases can be more toxic to the organism than the substance ingested. These transformations make modeling difficult, especially in organisms higher up the food chain.

Only two studies have presented a biokinetic model to describe the bioconcentration of organic UV filters. Gomez et al. [88] proposed a model to describe the bioconcentration of pharmaceutical and personal-care products (organic UV filters 2-ethyl-hexyl-4-trimethoxycinnamate (EHMC) and octocrylene (OC)) and benzodiazepines (diazepam and tetrazepam) in the marine mussel *Mytilus galloprovincialis*. Vidal-Liñán et al. [78] modeled the bioaccumulation kinetics of UV organic filters (4-methylbenzylidene camphor (4 MBC), benzophenone-3 (BP3), benzophenone-4 (BP4), OC, and octyl dimethyl p-aminobenzoate (ODPABA)) also using *M.galloprovinciallis*, suggesting that lipids are the final destination of the accumulated chemical substances.

However, sunscreens also contain trace metals and inorganic UV filters ($TiO_2$-NPs/$TiO_2$, ZnO-NPs/ZnO). The stability of inorganic filters in the aquatic medium is the subject of numerous studies, e.g., [72,89–91]. ZnO nanoparticles in the aquatic environment are transformed into their ionic form ($Zn^{2+}$) [92,93]. Both ZnO-NPs and $Zn^{2+}$ have been suggested to induce relevant toxicity effects, highlighting the release of $Zn^{2+}$ ions from ZnO-NPs as important contributors to toxic effects [22,94,95]. On the other hand, Fan et al. [96] modeled the bioconcentration of six different types of $TiO_2$-NPs in freshwater cladoceran *Daphnia magna*, observing significant effects on the exposure concentration and property of the nanoparticles, concluding that the studied nanoparticles could be considered as highly bioaccumulative.

Metals from sunscreen, once released into the aquatic environment, can bioconcentrate in organisms following the same processes as dissolved metals in the marine environment and, therefore, their bioconcentration can be described with models similar to those used in the bioaccumulation studies of dissolved metals obtained from sources other than sunscreen or just from synthetic solutions.

*Biokinetic and Toxicokinetic Models*

Bioaccumulation is an important framework in ecotoxicology research, serving as a link between the environment and organisms, being of crucial importance to assess the chemical status of water masses [97,98]. Any biological effect manifested by aquatic organisms due to a chemical is directly related to that dose at the site of action, and is the result of its bioaccumulation [97,99], which is modeled using biokinetic or toxicokinetic models.

Bio-toxicokinetic models apply the science and techniques of chemical kinetics to chemical behavior in biological systems [38]. The application of these models allows for

the mathematical characterization of the processes of chemical absorption, distribution, and elimination. Bio-toxicokinetic models incorporate bioconcentration through expressions based on the material equilibrium between influx and efflux rates through differential equations that describe their evolution over time [100]. They are compartmental models that describe the movement of toxic substances between compartments. A compartment represents the quantity of a compound that behaves as if it exists in a homogeneously well-mixed container, and moves across the compartment boundary with a single absorption- or elimination-rate coefficient [101]. Biokinetic models consider the organism as a whole (a single compartment). However, toxicokinetic models consider different compartments within the organism, providing information on the bioconcentration in each compartment and a prediction of the most affected organs and/or tissues [102–104]. Both biokinetic and toxicokinetic models are modeled using mass balances, generally considering that the mass transfer between compartments follows first-order kinetics [33,38,105].

Figure 4 schematically presented the processes that can be considered in the approach of these models. All the possible flows of the biokinetic models are presented in Figure 4a. The uptake of the substance by the organism can be performed through the gills, when the substance is dissolved or in a colloidal form, or through the digestive system if it is associated with particles or food. Once in the organism, it can bioconcentrate and finally be excreted into the aquatic environment.

Figure 4b illustrates the possible flows of matter considering a toxicokinetic model. The difference present in the biokinetic model is that it considers the flows of matter within the organism, the bioconcentration of the substance in the different compartments, and the interrelation between them.

Considering first-order kinetics, the bioconcentration of a substance in an aquatic organism can be modeled as biokinetic according to the following general expression:

$$\frac{d[C_t]}{dt} = k_u \cdot [C_W] + k_{u,col} \cdot [C_{Col}] + k'_u \cdot [C_{aq,int}] + (AE \cdot IR)_{food} [C_{food}] + (AE \cdot IR)_{sed} [C_{sed}] - k_e \cdot [C_t] \tag{1}$$

$[C_t]$ is the bioaccumulated concentration of the substance in the organism, $[C_W]$ the concentration of the substance in water, $[C_{Col}]$ is the concentration of the substance in a colloidal form, $[C_{aq,int}]$ is the concentration of the substance in the pore water of the sediment, $[C_{food}]$ is the concentration of the substance present in food, and $[C_{sed}]$ is the concentration of the substance in the sediment particles. $k_u$, $k_{u,col}$, and $k'_u$ represent the uptake-rate coefficients of the substance from water, colloids, and interstitial water, respectively. $k_e$ is the efflux-rate coefficient. $(AE \cdot IR)$ is the product of the efficiency of food assimilation in the digestive system (AE) produced by the rate of the specific ingestion of matter (IR). The subscript "food" refers to food and the subscript "sed" to sediment particles. Equation (1) describes all the possible flows; however, depending on the study, this equation can be simplified. If the uptake from the sediments and colloids is not considered, Equation (1) is simplified into Equations (2) and (3), with food and food and sedimemt as the uptake inputs respectively:

$$\frac{d[C_t]}{dt} = k_u \cdot [C_W] + (AE \cdot IR)_{food} [C_{food}] - k_e \cdot [C_t] \tag{2}$$

$$\frac{d[C_t]}{dt} = k_u \cdot [C_W] + (AE \cdot IR)_{food} [C_{food}] + (AE \cdot IR)_{sed} [C_{sed}] - k_e \cdot [C_t] \tag{3}$$

When the substance is dissolved in water, its uptake by food can be neglected and this term in Equation (2) can be considered as negligible, obtaining the following expression presented in Equation (4):

$$\frac{d[C_t]}{dt} = k_u \cdot [C_W] - k_e \cdot [C_t] \tag{4}$$

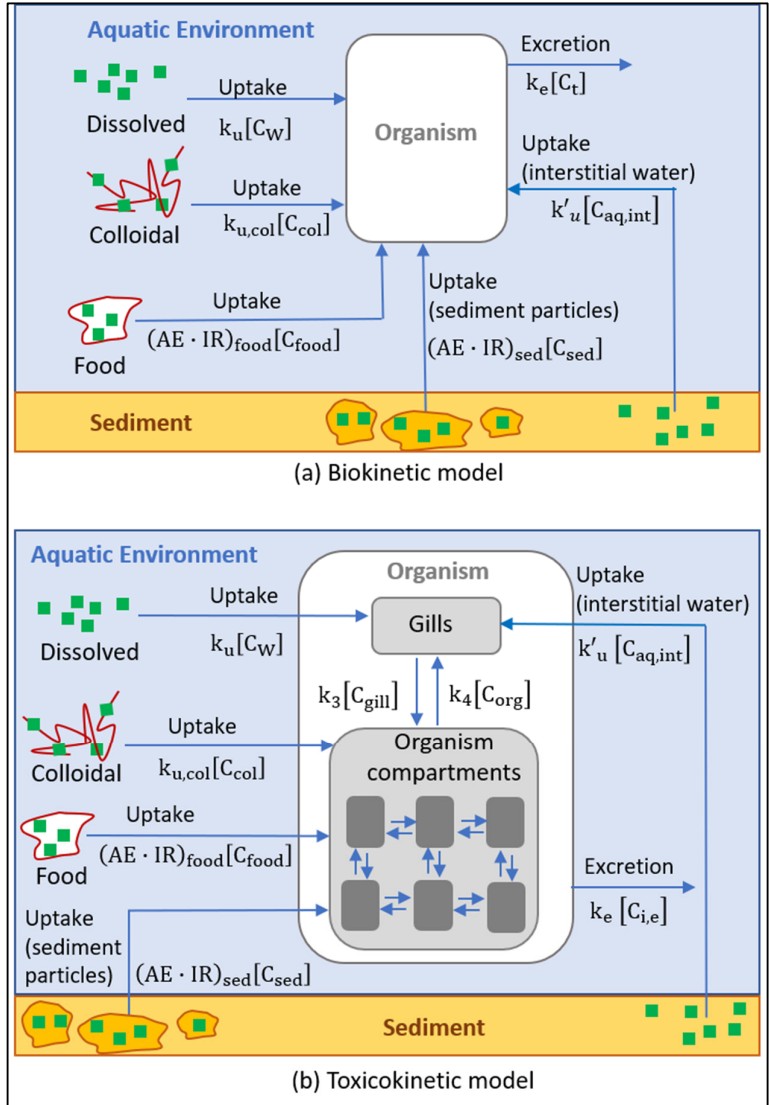

**Figure 4.** Processes of uptake, bioconcentration, and excretion of a substance by the aquatic organism. (**a**) Biokinetic model and (**b**) toxicokinetic model. Where $[C_t]$, $[C_W]$, $[C_{Col}]$, $[C_{aq,int}]$, $[C_{food}]$, $[C_{sed}]$, $[C_{gill}]$, $[C_{org}]$, and $[C_{i,e}]$ are the concentrations of bioaccumulated substances in water, in colloid form, in the pore water of the sediment, in the food, in the sediment particles, uptake by the gills, and by the digestive canal and excreted, respectively. $k_u$, $k_{u,col}$, $k'_u$, $k_3$, and $k_4$ represent the uptake-rate coefficients of the substance from water, colloids, interstitial water, gills, and other compartments, respectively; $k_e$ is the efflux-rate coefficient; $(AE \cdot IR)$ is the product of the efficiency of food assimilation in the digestive system (AE) by the rate of specific ingestion of matter (IR); the subscript "food" refers to food and the subscript "sed" to sediment particles.

The formulations mentioned above assume no organism growth, which is a reasonable assumption during short laboratory experiments, but may be violated when the models are applied to predict accumulation in the field. When the size of the organism increases, "growth dilution" occurs, as the new tissue mass dilutes the toxicant concentration [101]. The efflux-rate coefficient, $k_e$, of a growing organism overestimates the actual elimination. The growth-rate coefficient (g) must be incorporated into the model to correct the value of the efflux-rate coefficient. Considering the growth, Equation (5) subsequently describes the bioaccumulated concentration of the substance in the organism:

$$\frac{d[C_t]}{dt} = k_u \cdot [C_W] + (AE \cdot IR)_{food}\,[C_{food}] - (k_e + g) \cdot [C_t] \tag{5}$$

Table 3 presents the different studies compiled from the bibliography that proposed the biokinetic models. Bivalves were the main organisms used in these studies because they are considered as excellent candidates that biomonitor coastal pollution [106], as they are filter feeders and exposed to high volumes of water to feed and breathe. In this process, the gills and other exchange surfaces are exposed to metals and other contaminants present in the environment, accumulating them in their soft tissues and shells [105,107].

The simplest model (Equation (4)) was proposed, mainly to describe the bioaccumulation kinetics of short laboratory experiments, since it neglects the effect of the growth of the organisms and does not consider the effect of food or the presence of sediment. The coefficients of the uptake and output rates were determined in order to obtain the bioconcentration factors of substances as organic UV filters [78,88]. In addition, it was also used as a rough estimate to serve as a starting point for more complex models [108].

**Table 3.** Biokinetic models used in different studies.

| | UV Filters/Metals | Water Type | Field/Laboratory | Reference |
|---|---|---|---|---|
| $\frac{d[C_t]}{dt} = k_u \cdot [C_W] - k_e \cdot [C_t]$ | | | | |
| Mussel | 4-MBC, BP3, BP4, OC, OD-PABA | Seawater | Laboratory | [78] |
| Fish | Cu | Fresh water | Laboratory | [108] |
| Mussel | EHMC | Seawater | Laboratory | [88] |
| Clams/mussels | Zn | Seawater | Field | [26] |
| $\frac{d[C_t]}{dt} = k_u \cdot [C_W] + (AE \cdot IR)_{food}\,[C_{food}] - (k_e + g) \cdot [C_t]$ | | | | |
| Oysters/mussels/clams | Ag, Cd, Cr, Cu, Ni, Pb, Ti, Zn | Seawater | Field | [106] |
| Mussel | As, Cd, Cr, Cu, Ni, Se, Pb, Zn | Seawater | Laboratory | [109] |
| Clams | Ag, As, Zn | Seawater | Laboratory | [110] |
| Mussel | Cd, Cr, Cu, Ni, Zn | Fresh water | Field | [111] |
| Scallop | Cd, Zn | Seawater | Laboratory | [112] |
| Mussel | Ag, Am, Cd, Co, Se, Zn | Seawater | Field | [113] |
| $\frac{d[C_t]}{dt} = k_u \cdot [C_W] + (AE \cdot IR)_{food}\,[C_{food}] + (AE \cdot IR)_{sed}\,[C_{sed}] - k_e \cdot [C_t]$ | | | | |
| Clams/prawns/whiting | Cd, Mn, Zn | Seawater | Laboratory | [114] |
| Oyster | Zn | Seawater | Laboratory | [115] |

Lengthy experiments conducted at field and laboratory levels need to incorporate the effect of the organism's growth (Equation (5)). The laboratory studies analyzed the effects of different variables, such as the change in salinity [109] or changes in organism allometry [112]. Field studies allowed the validation of laboratory results or predicted their bioconcentration and effects on organisms in the aquatic environment. Some studies incorporated sediment particles as an uptake pathway (Table 3) with the aim of analyzing the main uptake pathway in different organisms [114,115].

The bioconcentration of a substance in different compartments within the organism can be modeled using toxicokinetic models (Figure 4b). In these models, the uptake of the

substance can be condcuted in two ways. Dissolved substances are mainly captured by the gills and solid substances (food or particles) of the digestive canal. Once in the organism, the substance can be bioconcentrated in different interrelated compartments. Considering the first-order kinetics, Equations (6) and (7) describe all the possible flows:

$$\frac{d[C_{gill}]}{dt} = k_u \cdot [C_W] + k_4 [C_{org}] + k'_u \cdot [C_{aq,int}] - k_3 \cdot [C_{gill}] \tag{6}$$

$$\frac{d[C_{org}]}{dt} = k_3 \cdot [C_{braq}] + (AE \cdot IR)_{food} [C_{food}] + (AE \cdot IR)_{sed} [C_{sed}] \pm \sum \frac{dC_i}{dt} - k_e \cdot [C_{i,e}] \tag{7}$$

where $[C_{gill}]$ and $[C_{org}]$ are the concentrations taken up by the gills and digestive canal, respectively; $[C_i]$ is the concentration of the substance in each compartment, i, of the organism; and $[C_{i,e}]$ is the concentration of the substance from which the substance is excreted. Kinetic constants $k_3$ and $k_4$ represent the uptake-rate coefficients from the gills and other compartments, respectively.

The term $\sum \frac{dC_i}{dt}$ depends on the compartment scheme (number of compartments and the relationship between them). It represents the bioaccumulation of the substance in the compartment and depends on the input and output flows considered. Each flux is usually assimilated to a first-order kinetics. For the model proposal, as many additional equations as the number of compartments must be indicated. The resolution of the system of differential equations is solved through matrix developments and, generally, through numerical methods implemented in specific software.

Equations 6 and 7 can be simplified if the uptake from sediments and feed is not considered, obtaining the following expressions presented in Equations (8) and (9):

$$\frac{d[C_{gill}]}{dt} = k_u \cdot [C_W] + k_4 [C_{org}] - k_3 \cdot [C_{gill}] \tag{8}$$

$$\frac{d[C_{org}]}{dt} = k_3 \cdot [C_{braq}] \pm \sum \frac{dC_i}{dt} - k_e \cdot [C_{i,e}] \tag{9}$$

Sánchez-Marín et al. [105] proposed two different three- or four-compartment models to clarify the entry pathway (gills or digestive canal) and elimination of copper complexes with different contents of humic acid in mussels. Modeling demonstrated that the colloidal metal was available for uptake through the gut of marine mussels. Redeker et al. [116] proposed a five-compartment model (water, sediment, gut, central, and peripheral) to model the bioaccumulation of cadmium and zinc from water and sediment by the aquatic oligochaete, *Tubifex tubifex*. Simulations of different conditions showed that both dissolved and sediment-associated metal can be important sources of metal exposure for worms, and that the relative importance was highly dependent on the metal and exposure conditions, including the lability of metals in the sediment phase.

Chen and Liao [108] proposed a biokinetic model to understand the effect of dissolved copper on tilapia (*Oreochromis mossambicus*), a freshwater fish, when subjected to pulsed variations of the metal in the aquatic environment. It proposed a 10-compartment toxicokinetic model to predict the damage of the different organism' organs.

Increasing numbers of studies have been designed to examine the effects of UV filters, mainly $TiO_2$ and ZnO nanoparticles, on different aquatic organisms, ranging from bacteria to fish. These studies identified and evaluated the toxicity and ecotoxicological effects, through several biological responses, and bioaccumulation obtained mainly in laboratory-based experiments under different environmental conditions, e.g., [3,28,30,31,85,117]. However, as it has been highlighted in this section, a reduced number of studies proposed mathematical models to describe the interaction of sunscreen with living organisms, which is necessary to assess and predict impacts more accurately.

## 4. Conclusions

Sunscreen has become a product widely used by people all over the world, both in everyday life and associated with sun and beach tourism. Since its invention, it has evolved through different stages to achieve optimal protection against the entire UV spectrum with increasingly complex formulations. However, restrictions of the use of certain components, due to the impact produced by their release and transformation in an aquatic environment, is a driver for the design of environmentally friendly formulations.

The analysis and modeling of the release and behavior of sunscreen ingredients into the marine environment, including aquatic organisms, allowed us to predict the fate and levels of the chemicals under study, and therefore allowed us to predict the potential negative impacts on the aquatic environment. A large group of models studied the behavior of the particles and chemicals released into the water by hydrodynamic and kinetic models in the laboratory, while the knowledge of the fate of sunscreen components in the real environment is still limited. The laboratory-scale works analyzed and model: (i) the physical–chemical property behavior of sunscreens once they are incorporated into the aquatic environment, (ii) the photodegradation of sunscreen components, and (iii) the release kinetics of metals present in sunscreens and the formation of nutrients in the receiving aquatic environment. The direct photolysis of sunscreens is modeled as pseudo-first-order kinetics, while the indirect pathway by the reaction of sunscreen with reactive oxygen species is described as second-order kinetics.

Another significant group of works analyzed the effect of the sunscreen components on marine biota (toxicokinetic models), which caused the bioaccumulation of certain compounds in the organisms' tissues. These works considered that the processes of uptake and excretion of chemicals to and from organisms, as well as their transfer between different organs, followed a first-order kinetic. The proposed models differ in the following ways: (i) the chemical uptake and (ii) the number of internal compartments of the organism considered, ranging from a single compartment to several compartments, such as the gills, digestive system, and internal organs. The kinetic parameters were obtained from the models for each species and the chemical allowed us to monitor the pollution levels in coastal areas, as well as the quality of the ecosystem.

The knowledge of the mechanisms of release and behaviour of the ingredients in sunscreens is essential to understand and predict their potential detrimental effects on marine ecosystems, as well as to identify their most environmentally harmful ingredients. The modeling approach was a useful prediction tool for these substances and can be of great value to environmental managers and stakeholders. However, the complexity (multitude of ingredients) and diversity (according to brand or format) of the sunscreen matrix, and the multitude of compartments and environmental variables that comprise the marine ecosystem, limited the application of these models. In this sense, future investigations should focus on the development of models, which consider the aforementioned factors, in order to obtain predictions of the potential impacts of these emerging products, and to inform future effective coastal-protection strategies and conservation policies based on sustainable tourism.

**Author Contributions:** All of the authors contributed to the study's conceptualization and successive reviews. G.R.-G. and J.R.V.F.: writing—original draft preparation and supervision. A.R.-R. and A.T.-S.: writing—review and editing. A.R.-R.: funding acquisition and project administration. All authors have read and agreed to the published version of the manuscript.

**Funding:** This work is funded by the 2014–2020 ERDF Operational Programme and by the Department of Economy, Knowledge, Business and University of the Regional Government of Andalusia (project reference: FEDER-UCA18-106672).

**Institutional Review Board Statement:** Not applicable.

**Informed Consent Statement:** Not applicable.

**Data Availability Statement:** Not applicable.

**Acknowledgments:** A. Rodríguez-Romero was supported by the Spanish grant Juan de la Cierva Incorporación referenced as IJC2018-037545-I.

**Conflicts of Interest:** The authors declare no conflict of interest.

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
