# Peer review of "Analysis and Modeling of Sunscreen Ingredients’ Behavior in an Aquatic Environment"

_2673-1924, doi:10.3390/oceans3030024_

Round 1

Reviewer 1 Report

The manuscript analyzes the behavior of the various ingredients of sunscreens in the aquatic environment with special attention to their modeling. In the introduction, some information is given on the manufacture, sale and current legislation on the subject. Then, the classification of sunscreens into two main groups is presented: organic and inorganic sunscreens. Then, the behavior of sunscreens in aquatic systems in the presence and absence of living organisms is studied. In the first case, a series of works carried out both under controlled laboratory conditions and in the field are analyzed, referring to the degradation and aging processes in an aquatic environment. In the second case, the mechanisms and mathematical models available in the literature to analyze and predict the release of these compounds, their transformation and their interactions with living aquatic organisms are described. The work ends with a synthesis of the different bio-toxicokinetic models.

The review paper is clear and well written, and a complete list of references is given (118 papers). However, there is no mention of how the review was conducted (the database used, the time frame of the papers studied, the inclusion and exclusion criteria, etc.).

Searching the Web of Science for the word "sunscreen" and filtering the results for "review articles" from "2018" in the fields of "environmental sciences," "toxicology," "marine freshwater biology," "ecology," "biodiversity conservation" yields a list of 35 papers. I suggest that the authors indicate how their review can be considered relevant and of interest to the scientific community.  I think the authors should do a better job of highlighting the overall benefits of publishing this work. Does the work advance the current state of knowledge? 

The figures are understandable, but I suggest indicating in the caption of Figure 4 where the description of all abbreviations shown in the figure can be found.

The conclusions are generally supported by the citations listed.

I think the paper can be accepted in principle after minor revisions based on the reviewer's comments.

Reviewer 2 Report

Review Comments of “Analysis and modelling of the sunscreen ingredients behaviour in the aquatic environment” Submitted Manuscript (ID: oceans-1744362).

Type of manuscript: Review

The submitted review paper evaluates the literature relative to analysis and modeling of sunscreen behavior in aquatic ecosystems. In particular, the review is divided into 2 main areas: 1) behavior of chemicals in aquatic environments, without considering organism presence; 2) behavior of chemicals in presence of organisms. The review consider several aspects of the issue related to sunscreens, including socio-economic ones, manufacturing aspects and, of course, environmental impact. About the latter, main reaction pathways of sunscreen are taken into account, including photo-degradation, oxidation, release of nutrients and dynamic models for transport in aquatic environments, dividing literature between organic and inorganic compounds. Moreover, also interactions with aquatic organisms were considered, including bioaccumulation, metabolism and bio- and toxico-kinetic pathways, again dividing between organic and inorganic compounds.

The entire review is well written, fluent and easy to read and to understand. Despite the topic is well known by scientific community, and several reviews on the same contaminants are findable with main search engines (ScienceDirect, WoS, Scopus, PubMed), this particular review adds a different point of view on this issue, investigating on chemical behavior of sunscreen compounds depending on the experimental treatment performed (in-lab, in-field, with organisms, without organisms), reporting also the necessity for more precise and realistic modelling of this kind, and in general for all the emerging contaminants. For these reasons I feel confident in considering the review acceptable for Oceans Journal after minor revisions.

Specific comments:

Line 37-39: skincare or skin care, be consistent all along the manuscript.

Line 77: Explain better what you mean with degradation products. Sunscreen, as well as all cosmetics, are considered to arrive in the environment almost unmodified. Are you intending chemical and/or photo- degradation products? Probably it may be useful to clarify, in order to avoid misunderstandings.

Line 497: species names go without uppercase letters (in the specific case it should be Daphnia magna). It’s surely a typo, please check all along the manuscript.

Reference List: please check all author name spelling, there are some mistaken names (e.g. 94 - Vieira Sanchez should be Vieira Sanches).

Reviewer 3 Report

This manuscript reviews the analysis and modelling of UV filters in different aquatic environments. Generally speaking, the MS is interesting, well organized, and the content is worth to be published. Only minor changes are necessary:

 -          The authors may consider to include the word “review” on the title.

-        -           In section 1.1 or 1.2, it would be nice to include a table highlighting the most common UV filters, their chemical and/or molecular formula, how many are regulated or banned, and the most important physical-chemical properties that should be taken in consideration for modelling.

-       -             In section 2.1, it would be nice to include a table similar to the one presented in section 2.2, highlighting the models. 

Round 2

Reviewer 2 Report

The authors well revised the entire manuscript taking into account all reviewers' suggestions. For that reason I feel comfortable indicating the publication as acceptable in the present form.

I'd like to suggest only a small check on Figure 1, which is not well visualized on the pdf, but that's probably an issue of pdf building process.

Reviewer 3 Report

The article is now suitable for publication.